# Effect of Seaweed-Based Biostimulants on Growth and Development of *Hydrangea paniculata* under Continuous or Periodic Drought Stress

**Paulien De Clercq** [1,2,*], **Els Pauwels** [2], **Seppe Top** [3], **Kathy Steppe** [3] **and Marie-Christine Van Labeke** [1,*]

[1] Department of Plants and Crops, Faculty of Bioscience Engineering, Ghent University, 9000 Ghent, Belgium
[2] Ornamental Plant Research (PCS, Proefcentrum voor Sierteelt), 9070 Destelbergen, Belgium
[3] Laboratory of Plant Ecology, Department of Plants and Crops, Faculty of Bioscience Engineering, Ghent University, 9000 Ghent, Belgium
[*] Correspondence: paulien.declercq@ugent.be (P.D.C.); mariechristine.vanlabeke@ugent.be (M.-C.V.L.)

**Abstract:** To adapt to climate change and water scarcity during dry, hot summers, more sustainable, or even deficit, irrigation is required in the ornamental sector, as it uses large amounts of water to sustain high-value crop production. Biostimulants, especially seaweed extracts, could offer a sustainable solution against drought stress as they are known to increase plant tolerance to abiotic stress. The effect of four seaweed extracts based on *Ascophyllum nodosum*, *Soliera chordalis*, *Ecklonia maxima*, and *Saccharina latissima* and one microbial biostimulant were tested on container-grown *Hydrangea paniculata* under drought stress conditions for two years. During the first trial year, in 2019, overall irrigation was reduced by 20%. In 2021, plants were subjected to repeated drying and wetting cycles. In general, less irrigation, and thus a lower substrate moisture content, reduced stomatal conductance, biomass production, and root development, but increased plant compactness. The biostimulants showed minor effects, but these were not observed in both experiments. Treatment with the *A. nodosum* extract resulted in longer branches and more biomass under deficit irrigation but tended to accelerate flowering when repeated drying and wetting cycles were applied. The *E. maxima* extract negatively affected the branching of *Hydrangea* under repeated drying and wetting cycles.

**Keywords:** woody ornamentals; biostimulants; seaweed extracts; drought stress; pigments; stress metabolites; reflectance; stomata; dendrometer (LVDT)

## 1. Introduction

Ornamental horticulture is a small, but economically important, sector within agriculture in Belgium. The sector had a production value of 511 million euros in 2020, demonstrating its economic importance. To obtain high-quality plants, the hardy nursery sector uses large amounts of water [1,2] but, due to climate change, growers are increasingly facing periods of prolonged drought and heat waves, from which legal restrictions on water use and water shortages can arise. In the future, growers will be forced to use less water in their growing system, such that plants could suffer from drought stress [3].

Plant biostimulants are defined in the European Regulation (EC) No 2019/1009 for fertilizing products as follows: 'A plant biostimulant shall be an EU fertilizing product the function of which is to stimulate plant nutrition processes independently of the product's nutrient content with the sole aim of improving one or more of the following characteristics of the plant or the plant rhizosphere: (a) nutrient use efficiency, (b) tolerance to abiotic stress, (c) quality traits, or (d) availability of confined nutrients in the soil or rhizosphere' [4]. Biostimulants consist of a variety of ingredients and formulations and therefore can be classified into different groups, e.g., humic and fulvic acids, protein hydrolysates and

other N-containing compounds, seaweed extracts and botanicals, chitosan and other polymers, inorganic compounds, beneficial fungi, and beneficial bacteria [5–7].

Since antiquity, seaweeds have been used in agriculture as a source of organic matter and fertilizer, but their biostimulant effects have only been discovered recently [6,8]. Seaweed extracts contain polysaccharides, e.g., laminarin, alginates, carrageenans, micro- and macronutrients, sterols, N-containing compounds such as betaines, and hormones as potentially bio-active components and can act on soils and plants. They can affect the physical, chemical, and biological properties of the soil, e.g., improvement of moisture-holding capacity and soil aeration, contribution to the fixation and exchange of cations, and promotion of beneficial soil microbes [6,8,9]. They may also affect root architecture by improving lateral root formation and increasing the total root volume thus facilitating the efficient uptake of nutrients and water [9,10]. Seaweed extracts influence photosynthesis through a reduced degradation of chlorophyll, possibly caused by betaines [11]. Hormones present in seaweed extracts, e.g., auxins, abscisic acid, gibberellins, and other classes of hormone-like compounds, are considered to be the major causes of biostimulant activity on crops. The hormonal effects may affect seed germination, plant establishment, and further growth and development [8,9]. Wally et al. (2013) found evidence that the hormonal effects of the brown seaweed *A. nodosum* are, to a lesser extent, related to the hormonal content of the seaweed extracts themselves, but are mainly linked with the up- and down-regulation of hormone biosynthetic genes in the plant tissues [12]. Furthermore, seaweed concentrates trigger early flowering and fruit set in several crop plants probably by initiating robust plant growth [9]. Finally, seaweed extracts have also been shown to alleviate a variety of abiotic stresses including drought, salinity, and nutrient stresses [13]. Many abiotic stress factors manifest as osmotic stress and cause secondary effects, such as oxidative stress, which will lead to an accumulation of reactive oxygen species (ROS). These are known to damage DNA, lipids, carbohydrates, and proteins and cause aberrant cell signaling [9,14]. The mode of action of seaweed extracts in alleviating abiotic stress is not well understood, but the presence of bioactive molecules in the extracts, such as betaines [15] and cytokinins [16], may play a role. Seaweed extracts also increase endogenous concentrations of stress-related molecules in treated plants, such as cytokinins, proline, antioxidants, and antioxidant enzymes [13].

In the current study, biostimulants based on different seaweed species were selected as they are well known to increase the drought tolerance of plants. The effects of three commercial biostimulants based on the seaweeds *A. nodosum*, *E. maxima*, and *S. chordalis*, and one experimental biostimulant based on *S. latissima*, are studied on *Hydrangea paniculata* grown under (a) deficit irrigation or (b) repeated drying and wetting cycles. A microbial biostimulant was also included as they are known to protect plants from adverse environmental conditions. We used specific plant monitoring tools, physiological plant parameters, and ornamental value to find the best biostimulant that allows reduced irrigation without a loss of the ornamental quality of *Hydrangea paniculata*.

## 2. Materials and Methods

### 2.1. Plant Material and Growing Conditions

This research was conducted in greenhouses at the Ornamental Plant Research station (Proefcentrum voor Sierteelt PCS, Destelbergen, Belgium), Destelbergen, Belgium (51°3′ N; 3°48′ E) during the growing seasons of 2019 and 2021.

In spring 2019, rooted plug plants of *Hydrangea paniculata* 'Phantom' were transplanted in 1.5 dm³ containers filled with a commercial peat-based substrate (Agaris, Belgium), supplemented with a controlled-release fertilizer (3 kg m⁻³ Osmocote® Exact 5/6 M 15-9-12 + 2 MgO + 1 kg m⁻³ media trace elements). In spring 2021, rooted stem cuttings of *Hydrangea paniculata* 'Little Alf' were transplanted in 3 dm³ containers with a commercial peat/coconut substrate (Agaris, Belgium), containing a PG-mix fertilizer for the first three months of growth (0.4 kg m⁻³, NPK 14-16-18 + trace elements; Agaris, Belgium). The

experiments were set up in a greenhouse (average air temperature of 18–20 °C; average relative humidity of 71–82%) to avoid interference with rainfall. Prior to the experimental setup, plants were uniformly irrigated according to good horticultural practices. The young plants acclimated 4–12 weeks, and were then randomly assigned to an irrigation-biostimulant treatment (control or drought, with or without biostimulants).

## 2.2. Experiment 1—Deficit Irrigation (2019)

In 2019, a total of five treatments were studied. Two irrigation treatments started four weeks after transplanting (13 June 2019). The control treatment, based on growers' advice, received 3 L m$^{-2}$ overhead irrigation (standard irrigation), while the treatment with a deficit irrigation (80%) received 2.4 L m$^{-2}$ (deficit irrigation). A reduction of 20% was chosen, as 10% reduction, tested in a preliminary screening, had minor effects on plant growth and stress level, and no biostimulant effects were observed under these growing conditions. Irrigation frequency was controlled by radiation sum when a threshold value of 20 MJ m$^{-2}$ was exceeded.

Only the plants grown under deficit irrigation were treated with a foliar spray of biostimulants (three biostimulant treatments). Three commercial seaweed extracts, one based on the seaweed *Ascophyllum nodosum* (Phylgreen; Tradecorp), one on *Soliera chordalis* (SeaMelPure; Olmix), and another based on *Ecklonia maxima* (Kelpak; Kelp Products International), were tested, in comparison with a non-treated deficit control (DS Control) (Table 1). The doses and frequencies were specified by the manufacturer (Table 1).

All five treatments were repeated four times in a randomized block design. An experimental unit consisted of nine measuring plants surrounded by eleven border plants distributed in seven trays. In total, 400 plants were present. This experiment was completed at the end of October 2019.

## 2.3. Experiment 2—Repeated Drying and Wetting Cycles (2021)

In 2021, plants were irrigated by a drip irrigation system, one dripper per plant. The supply was set at 250 mL per dripper, corresponding to an irrigation of 0.88 L m$^{-2}$. The irrigation frequency was controlled by radiation sum set at 8 MJ m$^{-2}$, so plants were watered one to three times a day during the summer season. During the summer months, three drying cycles were applied by turning off the irrigation, followed by a recovery period compared with a continuously well-irrigated control so that two irrigation treatments were present. The first drying cycle in June started 12 weeks after transplanting (23 June–1 July) and ended before the presence of wilting symptoms, due to high temperatures. During the second cycle in July (16–22 July) and the third and final cycle in August (19–25 August), the plants were kept under water deprivation until they showed wilting symptoms. Measurements were only performed during the first and last drying cycle. The trial ended at the end of the growing season in September.

In 2021, two commercial seaweeds applied in 2019 were tested again, namely the *E. maxima* extract and the *A. nodosum* extract, the latter in combination with the application of a biostimulant based on hydrolyzed proteins (Delfan Plus V; Tradecorp). In addition, an experimental seaweed extract based on *Saccharina latissima* (North Sea Farmers) and a commercial biostimulant based on micro-organisms (Previsan S; Agriton) were included (Table 1). Biostimulants (four treatments) were tested on plants grown under repeated drying and wetting cycles compared with a non-treated control (DS Control). No biostimulants were applied to the continuously well-irrigated plants (No stress Control). A total of six treatments were present, which were repeated four times in a randomized block design. An experimental unit consisted of ten measuring plants and twelve border plants. A total of 288 plants were present.

**Table 1.** Application doses and frequencies of the tested biostimulants.

| Trial Year | Biostimulant | Application Dose | Application Frequency |
|---|---|---|---|

| | | | |
|---|---|---|---|
| 2019 | *Ascophyllum nodosum* extract (Phylgreen, Tradecorp) | 1.5 L ha⁻¹ | Every 15 days |
| | *Soliera chordalis* extract (SeaMelPure, Olmix) | 2 L ha⁻¹ | One application two weeks after planting |
| | *Ecklonia maxima* extract (Kelpak, Kelp Products International) | 2.5 L ha⁻¹ | First application: 7–10 days after planting; repeated at 14–21 days intervals up to four applications |
| 2021 | *Ascophyllum nodosum* extract combined with product based on plant-based amino acids (Phylgreen + Delfan Plus V, Tradecorp) | Phylgreen: 0.5 mL L⁻¹ Delfan Plus V: 2 mL L⁻¹ | Phylgreen: Every 15 days Delfan Plus V: during stress |
| | *Ecklonia maxima* extract (Kelpak, Kelp Products International) | 2.5 L ha⁻¹ | First application: 7–10 days after planting; repeated at 14–21 days intervals up to four applications |
| | *Saccharina latissima* extract (experimental product) | 3 mL L⁻¹ | Every two weeks |
| | Previsan S (Agriton) | 30 mL L⁻¹ | Every two weeks |

*2.4. Substrate Moisture Content Measurements*

Substrate characteristics (volumetric moisture content, electrical conductivity, and temperature) were determined using a WET sensor (Delta-T Devices Ltd., Cambridge, UK). During the deficit irrigation trial, measurements were performed every two weeks since continuous drought stress was expected, resulting in twelve measurements. During the drying–wetting cycle treatment, measurements were performed before the irrigation stop, (almost) every day of the drought period (no irrigation), and after drought recovery of three–four days. A total of eight (Exp. 2, cycle 1) or sixteen (Exp. 2, cycle 3) determinations of volumetric moisture content were performed.

*2.5. Plant Physiological Responses*

Chlorophyll and flavonoid levels in the leaves were determined non-destructively using a DUALEX® (Force A, Orsay, France). Hyperspectral reflectance spectra were determined at leaf level with a PolyPen RP410 (Photon Systems Instruments, Drásov, Czech Republic). Based on the hyperspectral data, selected indices were calculated (Table 2). The red edge inflection point (REIP) was calculated by determining the maximum value of the first deviation of the hyperspectral curve in the red region. A total of four measurements, on the two youngest fully developed leaves of two measurement plants were taken per experimental unit ($n = 16$). Twice as many measurements were performed in the second experiment in 2021.

**Table 2.** Reflectance indices calculated by the PolyPen RP410 based on hyperspectral data.

| Reflectance Index | Formula | Reference |
|---|---|---|
| NDVI (Normalized Difference Vegetation Index) | $NDVI = \dfrac{R_{NIR} - R_{RED}}{R_{NIR} + R_{RED}}$ | Ref. [17] |
| Lic1 (Lichtenthaler Index 1) | $Lic1 = \dfrac{R_{790} - E_{680}}{R_{790} + R_{680}}$ | Ref. [18] |
| Ctr2 (Carter Index 2) | $Ctr2 = \dfrac{R_{695}}{R_{760}}$ | Ref. [19] |
| ARI1 (Anthocyanin Reflectance Index 1) | $ARI1 = \dfrac{1}{R_{550}} - \dfrac{1}{R_{700}}$ | Ref. [20] |

Chlorophyll fluorescence was measured using a MINI-PAM II (Walz, Effeltrich, Germany). After 20–30 min of dark adaptation, the initial fluorescence ($F_0$) was determined, followed by a saturating flash (>4000 µmol m$^{-2}$ s$^{-1}$; 8 s) to determine the maximum fluorescence level ($F_M$). The maximum quantum efficiency of PSII ($F_V/F_M$) was calculated as the ratio of the difference between $F_M$ and $F_0$ over $F_M$ [21].

Stomatal conductance was measured five hours after sunrise, around midday (between 11:00 a.m.–01:00 p.m.), using a porometer (Delta-T Devices Ltd., Cambridge, UK). A total of eight (Exp. 1) and sixteen (Exp. 2) replicates per treatment were obtained by measuring one leaf, the youngest fully developed, per plant and two plants per experimental unit each time.

During the last drying cycle of experiment two, continuous measurements of stem diameter variation using linear variable displacement transducers (LVDT; DF series, Solartron Metrology Ltd., Steyning Way, UK) were performed on fifteen plants to monitor one to three repetitions in all six treatments. The data obtained from the LVDTs result in variations in stem diameter thickness (mm) after calibration. Calibration was undertaken beforehand giving a linear regression with $R^2 > 0.998$ for each sensor. The continuous stem diameter variation data were used to calculate daily stem diameter growth (the difference between stem thickness at midnight between two consecutive days) and stem shrinkage (the difference between the thickest and smallest stem diameter during that day). These calculations were performed every day during the observed period. Three sensors per treatment were installed but due to the movement of the stems during growth, erroneous displacement of the sensor head could occur, resulting in fewer replicates.

*2.6. Morphological Parameters*

At the end of both experiments, the number of branches was counted, and the length of the longest branch was measured. The plants were then harvested by cutting the stems just above the substrate to determine fresh and dry weight. The latter was performed by heating the above-ground biomass at 70–90 °C for at least 48 h. Finally, substrates were removed from their containers to examine the visible root distribution on a relative scale from 1 to 5 (Exp 1): 1—almost no visible roots, 2—limited visible roots at the bottom, 3—well-developed roots, but not all around the pot, 4—good root development all around the pot, and 5—excellent rooting (Figure S1 in Supplementary Materials). For the second experiment, the containers were double in volume and roots were in general less developed, so a slightly different scale was used: 1—almost no visible roots, 2—limited visible roots at the side, 3—limited root development at both the side and the bottom, 4—root development all around the pot and limited at the bottom, 5—good root development all around the pot and at the bottom (Figure S2). In experiment 1, six plants were harvested from each experimental unit (n = 24). In experiment 2, fourteen plants were harvested from each experimental unit, and thus 56 plants per treatment. Because of the late pinching to

stimulate branching in the first experiment, inflorescences could not be assessed. At the end of the second trial, the number of inflorescences was counted, and the development of each inflorescence was divided into five categories: 1—closed bud, 2—first elongation of the inflorescence with flower clusters still together, 3—second elongation of the inflorescence with the extension of the green flower clusters, 4—flowers open but still green, and 5—whitening of the flower (Figure S3).

### 2.7. Statistical Analysis

Statistical analysis was performed using Rstudio (R version 4.0.2) [22], completed with packages for specific statistical tests and making graphs [23–32]. First, data were checked on the presence of outliers. If the data complied with normality and homoscedasticity, results were subjected to a two-way analysis of variance with treatment and block as main effects (ANOVA). As no block effects were observed, the main effects were further analyzed by a post-hoc Tukey HSD test ($p \leq 0.05$). For the comparison of well-watered and drought stress without biostimulants, a Student's *t*-Test was performed. Non-parametrical data were analyzed by a Scheirer–Ray–Hare test [33], a non-parametrical alternative for a two-way ANOVA and extension of the Kruskal–Wallis test, followed by a post hoc Dunn's test with 'Benjamini-Hochberg' correction ($p \leq 0.05$) in case of comparison of multiple treatments. Two treatments were compared using a Mann–Whitney U test. All results were expressed as means ± Standard Error (SE).

## 3. Results

### 3.1. Effect of Biostimulants under Deficit Irrigation in 2019

#### 3.1.1. Effect of Deficit Irrigation

Deficit irrigation had a significant effect on most of the soil- and plant-related parameters determined during the first experiment (Table 3). The volumetric water content of the substrate, measured twice weekly to evaluate the effect of reduced irrigation, was on average 27 vol% under the standard irrigation, but was significantly lower under deficit irrigation. When less water was available in the substrate, a significantly lower electrical conductivity was observed. The Dualex and PolyPen RP410 were used to indirectly determine the effect of reduced irrigation on pigment and secondary metabolite contents. Chlorophyll seemed to be concentrated in the leaves of plants under deficit irrigation, as a significantly higher chlorophyll index was measured. Furthermore, a slight but significant increase in REIP (Red Edge Inflection Point) compared with standard irrigation was noted. The maximum quantum efficiency ($F_V/F_M$) was measured to evaluate the effect of reduced irrigation on photosystem 2. No significant effects were observed. To investigate the effect of reduced irrigation on stomatal conductance, measurements with the porometer were performed. Due to the lower substrate moisture content, a significantly lower stomatal conductance was measured. Table 3 shows that a reduced irrigation supply also had significant effects on the morphological parameters linked to the plant growth and quality of *Hydrangea*. The plants under deficit irrigation showed a higher branching degree but the branch length was reduced by 50%. Consequently, fresh and dry weights were reduced and these plants showed a less developed root system.

**Table 3.** Effect of deficit irrigation (trial 2019) on the substrate- and plant-related parameters of *Hydrangea* compared with a standard irrigation treatment. The average of each parameter over the trial is presented ± SE. Different letters (a and b) per parameter indicate a significant difference at $p \leq 0.05$.

| Parameter | Standard Irrigation (100%) | Deficit Irrigation (80%) (DS Control) | Statistics |
|---|---|---|---|
| **Substrate-related parameters** | | | |
| Volumetric moisture content [vol%] | 26.6 ± 1.1 a | 20.9 ± 1.2 b | $p = 0.0003$ [1] |

| | | | | |
|---|---|---|---|---|
| EC [3] [mS.m$^{-1}$] | 221.3 ± 7.1 a | 178.6 ± 6.0 b | $p = 0.0005$ [1] | |
| **Plant-related parameters** | | | | |
| Chlorophyll index [-] | 22.25 ± 0.50 b | 25.00 ± 0.49 a | $p < 0.0001$ [1] | |
| Flavonol index [-] | 0.675 ± 0.019 a | 0.712 ± 0.026 a | $p = 0.2394$ [1] | |
| REIP [4] [nm] | 702.8 ± 0.9 b | 704.4 ± 0.8 a | $p = 0.0464$ [2] | |
| NDVI [5] [-] | 0.505 ± 0.005 a | 0.506 ± 0.004 a | $p = 0.8599$ [1] | |
| Ctr2 [6] [-] | 0.438 ± 0.006 a | 0.432 ± 0.004 a | $p = 0.9867$ [2] | |
| Lic1 [7] [-] | 0.573 ± 0.005 a | 0.576 ± 0.004 a | $p = 0.7523$ [2] | |
| ARI1 [8] [-] | 0.296 ± 0.012 a | 0.251 ± 0.014 b | $p = 0.0136$ [1] | |
| $F_V/F_M$ [-] | 0.794 ± 0.003 a | 0.784 ± 0.008 a | $p = 0.0685$ [1] | |
| Stomatal conductance [mmol m$^{-2}$ s$^{-1}$] | 239.8 ± 20.2 a | 132.1 ± 11.8 b | $p = 0.0011$ [2] | |
| Branch length [cm] | 63.1 ± 1.5 a | 31.5 ± 0.9 b | $p < 0.0001$ [2] | |
| Number of branches [-] | 9.8 ± 0.6 b | 11.6 ± 0.5 a | $p = 0.0167$ [1] | |
| Fresh weight [g] | 89.6 ± 4.3 a | 51.2 ± 2.5 b | $p < 0.0001$ [1] | |
| Dry weight [g] | 29.2 ± 1.3 a | 14.9 ± 0.7 b | $p < 0.0001$ [2] | |
| Water content [%] [(FW-DW)/FW × 100] | 67.3 ± 0.3 b | 70.7 ± 0.3 a | $p < 0.0001$ [2] | |
| Root development score [-] | 3.42 ± 0.10 a | 2.63 ± 0.2 b | $p = 0.0002$ [2] | |

[1] treatment effect by a two-way analysis of variance (ANOVA). [2] treatment effect by a Scheirer–Ray–Hare test. [3] Electrical conductivity. [4] Red Edge Inflection Point. [5] Normalized Difference Vegetation Index. [6] Carter Index 2. [7] Lichtenthaler Index 1. [8] Anthocyanin Reflectance Index 1.

### 3.1.2. Effect of Biostimulants

The seaweed-based biostimulants were given as foliar applications, meaning that no influence on the volumetric substrate moisture content was expected and observed. The biostimulants had a limited influence on the chlorophyll and secondary metabolite contents, non-destructively determined with the Dualex and the PolyPen RP410 (Table S1). After four weeks of deficit irrigation, treatment with *E. maxima* significantly increased the chlorophyll index compared with the other biostimulant treatments but was not significantly different from the DS Control (non-treated deficit irrigation) (+10.7%). This initial positive effect disappeared during the growing season. One-time differences were also recorded for other spectral indices determined with the PolyPen RP410. After four weeks of deficit irrigation, a significantly different ARI1 index (Anthocyanin Reflectance Index 1) was observed between treatment with *S. chordalis* and *E. maxima*, where the last treatment had a 27.5% higher index. At the end of the trial, in September, differences in the Ctr2 index (Carter Index 2) and the REIP were observed between treatment with *A. nodosum* and treatment with *E. maxima*. Differences in the Lic1 index (Lichtenthaler Index 1) were observed between treatment with *A. nodosum* (+12.4%) and *S. chordalis*. Although these effects were not different from the DS Control. Stomatal closure is an adaptation response to drought stress. The commercial biostimulant *A. nodosum* tended to increase on average the stomatal conductance (147.4 mmol m$^{-2}$ s$^{-1}$ ± 12.0, +11.6%) compared with the DS Control (132.1 mmol m$^{-2}$ s$^{-1}$ ± 11.9), but this effect was not significant. The other biostimulants resulted on average in a slightly lower stomatal conductance, but again the differences were not significant (Table S2).

Figure 1 shows that the tested biostimulants did affect the morphological and plant quality parameters at the end of the trial. The *A. nodosum* treatment significantly increased the branch length by +27.9% but not the number of branches compared with the DS Control. The fresh and dry weights, as well as the root development, tended to increase for the plants treated with *A. nodosum* compared with the DS Control, though these effects were not significant. The water content was significantly lower for the plants treated with *A. nodosum* in comparison with the DS Control (−4.2%). The other tested biostimulants did not affect any of these parameters compared with the DS Control.

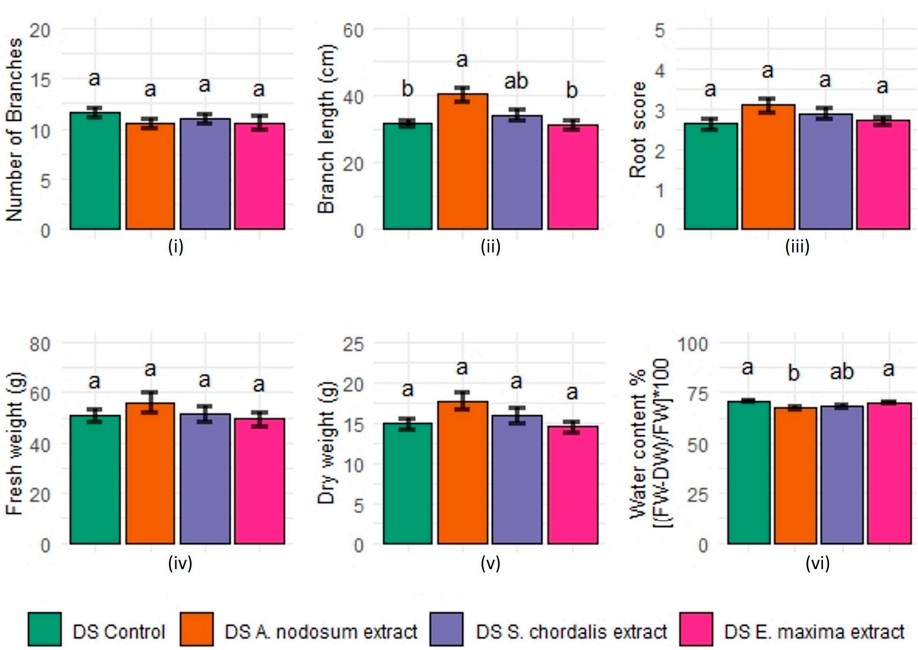

**Figure 1.** Effect of seaweed-based biostimulants on morphological parameters of *Hydrangea panicu-lata* under deficit irrigation (DS) (DS Control = non-treated deficit irrigation). (i) number of branches, (ii) length of the longest branch, (iii) root development score, (iv) fresh and (v) dry weight, (vi) plant water content. Fresh weight and water content were analyzed by a two-way ANOVA and a Tukey HSD test, the other parameters by a Scheirer–Ray–Hare test followed by a Dunn's test. Different letters (a and b) per parameter indicate a significant difference at $p \leq 0.05$ (Mean ± SE, $n = 24$).

### 3.2. Effect of Biostimulants under Repeated Drying and Wetting Cycles in 2021

In the second trial, irrigation was turned off three times during a hot period in June, July, and August, resulting in a substantial decrease in the volumetric moisture content of the substrate from ± 60 vol% to 25 vol% or lower during the following days, daily measured with the WET-sensor. When visible wilting started in the youngest leaves, irrigation was turned on again so plants could recover (Figure 2). Biostimulants were sprayed on the leaves, so no effects of the biostimulants on the volumetric moisture content of the substrate were expected.

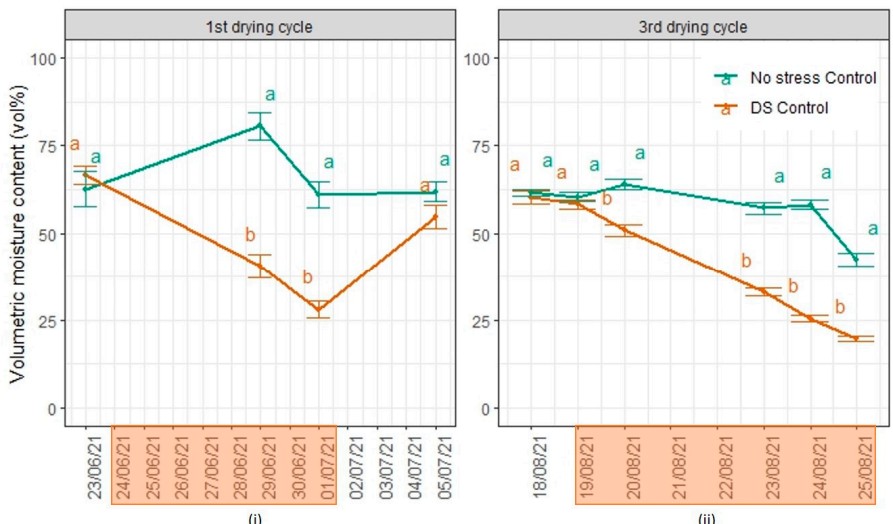

**Figure 2.** Effect of the drying cycles (DS Control) on the volumetric moisture content of the substrate of *Hydrangea* compared with optimal irrigation conditions (No stress Control). (i) first cycle: 23 June

2021–1 July 2021, (ii) third cycle: 19–25 August 2021. Highlighted dates indicate stress period. Treatments at 29 June 21, 1 July 2021, 5 July 2021, 19 August 2021, and 20 August 2021 were compared by a two-way ANOVA. Results on the other measurement days were analyzed by a Scheirer–Ray–Hare test. Different letters (a and b) between the treatments per measurement day indicate a significant difference at $p \leq 0.05$ (Mean ± SE, *n_cycle1* = 8, *n_cycle3* = 16).

Few differences between irrigation and biostimulant treatments were detected in pigment and secondary metabolite contents during both measured drying cycles (Table S3). Water shortage significantly increased the chlorophyll index in the leaves compared with the no-stress treatment. The largest difference in chlorophyll index, determined with the Dualex, between the No stress Control treatment (24.07 ± 0.42) and the DS Control (29.84 ± 0.50; +24%) was observed at the end of the drying period. Furthermore, the stressed plants treated with biostimulants showed a significantly increased chlorophyll index compared with the No stress Control treatment; the effect of *A. nodosum* was less. Before the start of the first drying cycle, *E. maxima* resulted in a significant decrease in the flavonol index by 13.3% compared with the DS Control and a 15% decrease compared with the treatment with the *S. latissima* extract. During the drying period and after recovery, no differences in indices between treatments were observed. During the third drying cycle, there were also no effects of drought stress nor biostimulant treatment on the flavonol index, NDVI, Ctr2, Lic1, and REIP.

The effect of the drying cycles and the decreasing volumetric moisture content of the substrate was also reflected in the stomatal conductance of *Hydrangea* leaves measured with the porometer. The results in Figure 3 show that, in both measured periods, there was a significant reduction in stomatal conductance during the drying cycle compared with the continuously well-irrigated treatment, starting from a substrate moisture content below 25 vol%. Furthermore, the influence of the biostimulants on the stomatal closure was investigated. Before the first irrigation stop (23 June 2021), the plants treated with Previsan S already had a significantly lower stomatal conductance compared with the DS Control and the plants treated with *E. maxima*. This difference disappeared during the drying period. Treatment with *E. maxima* tended to increase stomatal conductance during drought but this effect was not significant and also not present during the third drying cycle. Plants treated with the *S. latissima* extract had a significantly lower stomatal conductance compared with the No stress Control treatment. During the third drying cycle, no significant effects of the biostimulants on the decreasing stomatal conductance were observed.

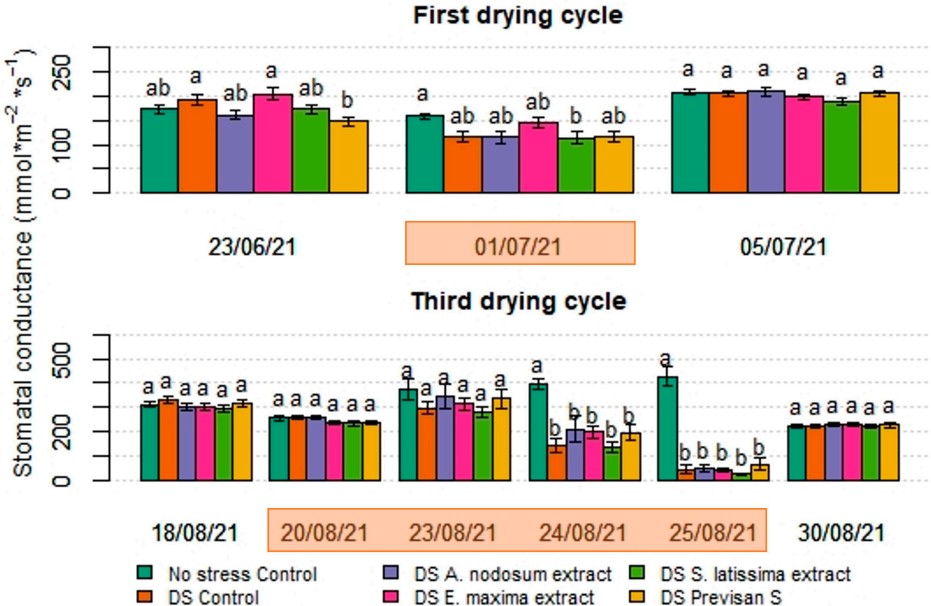

**Figure 3.** Effect of drying cycles without (DS Control) and with biostimulant treatments on the stomatal conductance of *Hydrangea*, compared with optimal irrigation (No stress Control). Highlighted dates indicate stress period. Data were analyzed by a two-way ANOVA followed by a Tukey HSD test, except for data on 23 June 2021, 24 August 2021, and 25 May 2021. These results were analyzed by a Scheirer–Ray–Hare test followed by a Dunn's test. Different letters (a and b) per measurement day indicate significant differences at $p \leq 0.05$ (Mean ± SE, $n = 16$).

The diel variations in the stem thickness of the control plants and the biostimulant-treated plants were measured continuously with LVDT (linear variable displacement transducer) sensors during the third drying cycle (Figure 4). It was mainly stem diameter shrinkage that showed the effects of drought stress on the plants. The No stress Control had a uniform and expected shrinkage over the consecutive days as these plants did not suffer from water shortage. From 23 August onwards, the stem shrinkage of plants without irrigation started to increase, which is one day before stomata started to close. At the time the stomata were almost closed at 18 vol% moisture content in the substrate (25 August 2021), the stem shrinkage was the largest. Plants treated with the biostimulant *A. nodosum* appeared to reduce stem shrinkage, whereas treatment with Previsan S increased it. No significant differences were calculated, as the number of repetitions for some treatments was too limited. These effects on stem thickness, induced by *A. nodosum* and Previsan S, were not observed on stomatal conductivity measurements.

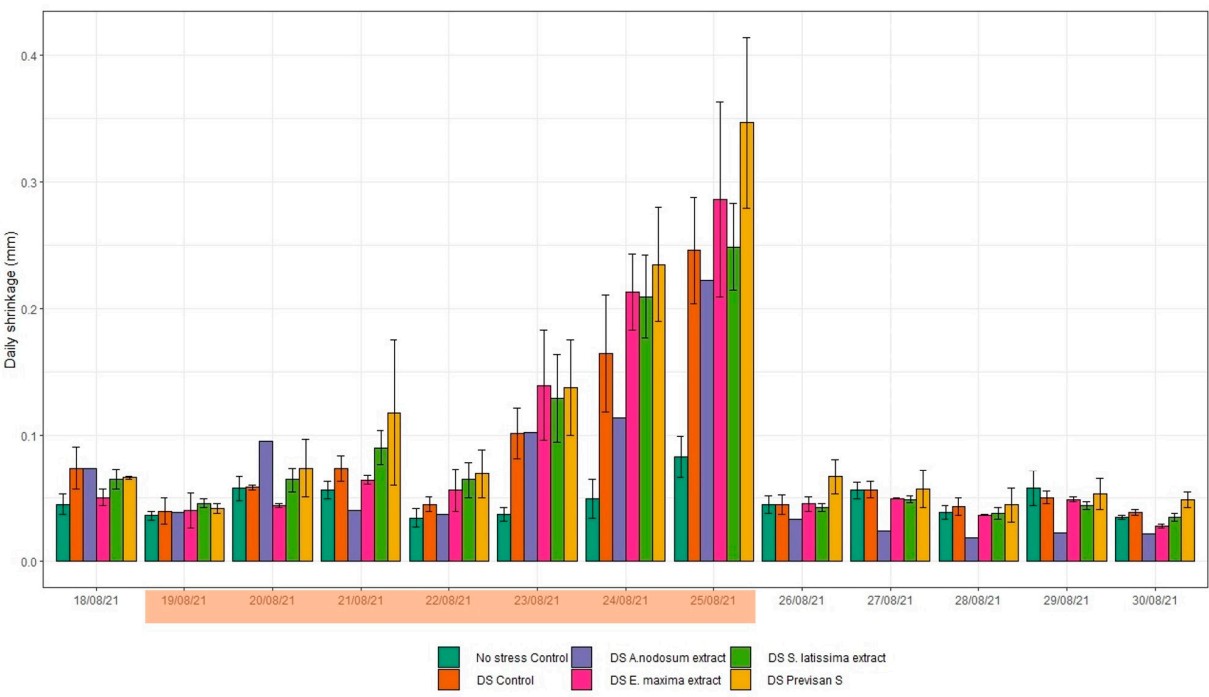

**Figure 4.** Effects of repeated drying cycles without (DS Control) and with biostimulant treatments on the daily stem shrinkage of *Hydrangea* before, during, and after the third drying cycle compared with optimal irrigation (No stress Control). Highlighted dates indicate stress period. (Mean ± SD (Standard Deviation*)); n_No stress Control* = 3, *n_DS Control* = 3, *n_DS A. nodosum extract* = 1, *n_DS E. maxima extract* = 2, *n_DS Previsan S* = 2).

At the end of the trial, the effects of repeated drying and wetting cycles and the biostimulant treatments on plant quality traits were evaluated. Figure 5 shows that the drying cycles with or without the application of biostimulants decreased the number of branches, but this was only significant for *E. maxima*. The repeated drying cycles with or without biostimulants significantly reduced branch length by 8.1% on average, though this was less pronounced for treatment with *E. maxima* (−5.5%). Water shortage negatively influenced root development compared with the No stress Control treatment. Here, the *A. nodosum* and *E. maxima* treatments improved the root development under stress conditions, up to the same root score as the No stress Control treatment. The lowest root score was obtained for the Previsan S treatment. The fresh and dry weight of the above-ground biomass decreased considerably due to the drying cycles by 24.7% and 24.2%, on average. Biostimulants had no additional effect.

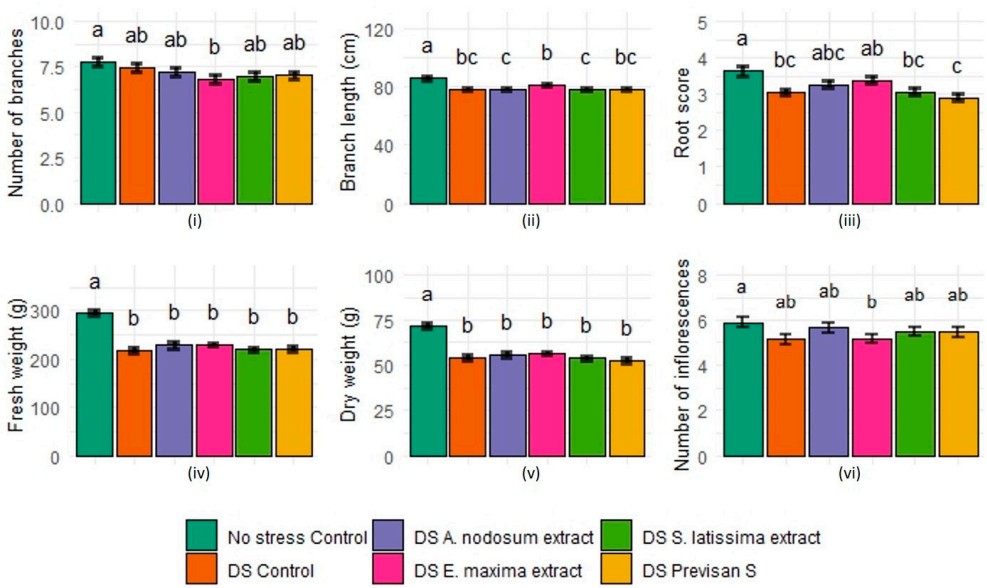

**Figure 5.** Effects of repeated drying cycles without (DS Control) and with biostimulants on plant quality traits compared with optimal irrigation (No stress Control): (i) number of branches, (ii) length of the longest branch, (iii) root development score, (iv) fresh and (v) dry weight, and (vi) total number of inflorescences. Results of branch length and dry weight were analyzed by a two-way ANOVA followed by a Tukey HSD test. Other results were analyzed by a Scheirer–Ray–Hare test combined with a Dunn's test. Different letters (a and b) per parameter indicate a significant difference at $p \leq 0.05$ (Mean ± SE; $n = 56$).

At the end of the trial, the number of inflorescences was also counted. Each inflorescence was divided into five inflorescence development stages. *Hydrangea paniculata* has terminal inflorescences, so the number of branches had a strong influence on their total number. Again, treatment with *E. maxima* resulted in significantly fewer inflorescences, as this treatment also resulted in fewer branches (Figure 5). The drying cycles did not accelerate flowering; the DS Control had even more inflorescences in the first development stage compared with the No stress Control. Treatment with biostimulants affected the development of the inflorescences. *A. nodosum*, *S. latissima* extract, and Previsan S showed a more advanced development with fewer inflorescences in the first developmental stage (score 1) compared with the DS Control. This effect was most pronounced for *A. nodosum* with significantly more fully developed inflorescences (score 5) compared with all other treatments (Figure 6).

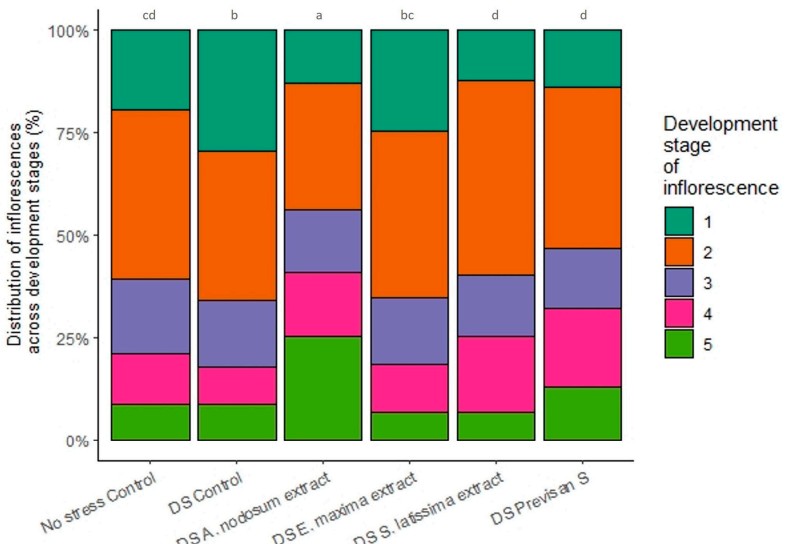

**Figure 6.** Effect of the drying cycles without (DS Control) and with biostimulants treatments on flower development of *Hydrangea paniculata* compared with optimal irrigation (No stress Control). Development of inflorescences was assessed using a 1–5 scoring system (1—closed bud; 2—first elongation of the inflorescence with flower clusters still together; 3—second elongation of the inflorescence with the extension of the green flower clusters; 4—flowers open but still green; 5—whitening of the flower). Differences in treatments on flower development were evaluated by a Scheirer–Ray–Hare test, followed by a post hoc Dunn's test. Different letters (a, b, c, and d) indicate a significant difference ($n = 56$).

## 4. Discussion

### 4.1. Effect of Deficit Irrigation and Repeated Drying Cycles on Hydrangea

Both deficit irrigation and repeated drying cycles significantly reduced the volumetric moisture content of the substrate compared with the well-watered (no stress) controls. This was accompanied by a significantly higher chlorophyll index measured in the plants under deficit irrigation compared with the standard irrigation. Furthermore, the REIP (red edge inflection point) shifted to higher wavelengths indicating a higher chlorophyll content [34]. The chlorophyll seemed to be more concentrated due to a lower leaf water content. Marenco et al. (2009) and Martínez and Guiamet (2004) also found a negative correlation between the leaf water content and the chlorophyll content measured with the SPAD-meter in different Amazonian tree species, wheat, and maize [35,36]. The production of secondary metabolites, especially phenylpropanoids such as flavonoids and flavonols, is induced by various biotic and abiotic environmental stresses [37]. Furthermore, oxidative and drought stress induced the increased the production of these secondary metabolites, including flavonols and anthocyanins, to mitigate the effects of stress with their strong radical scavenging activity [38,39]. In this research, no increased flavonol or anthocyanin content, determined by the optical sensors, was observed under reduced irrigation.

Moreover, no effect of reduced irrigation on maximal photochemical yields was observed. The photosynthetic system appears to remain intact above certain drought stress levels. Several studies reviewed by Flexas et al. (2004) show that the $F_V/F_M$-ratio remained constant as long as the stomatal conductance remained above 50 mmol m$^{-2}$ s$^{-1}$, the general threshold for severe drought stress in C3 plants. However, the $F_V/F_M$-ratio abruptly decreased at lower stomatal conductance, indicating a down-regulation of the entire photosynthetic metabolism at this stress level [40,41]. In the research of Liu et al. (2010) on different woody ornamental species, the same pattern was observed during repeated cycles of drying and rewetting [42].

This reduced irrigation went along with a lower stomatal conductance. From the moment the volumetric moisture content fell below 25 vol%, plants responded to the water

deficit by closing their stomata. Turner (1991) also observed stomatal response only from the moment a certain threshold of soil water content was exceeded [43]. The moment the moisture content was about 20 vol%, stomata were almost closed compared with the well-irrigated treatment. In *Hydrangea paniculata* and *Petunia x hybrida*, grown in another type of substrate, stomatal conductance was still between 300—500 mmol m$^{-2}$ s$^{-1}$ at a moisture content of 20–35 vol% [44,45]. The stomata of *Petunia x hybrida* almost closed when the plant experienced severe drought (moisture content around 10 vol%) [45]. Flexas and Medrano (2002) [40,46] defined four different phases of drought based on the daily maximum stomatal conductance ($g_s$) of different C3 crops (mild drought stress: $g_s$ > 150 mmol m$^{-2}$ s$^{-1}$; moderate drought: 150 mmol m$^{-2}$ s$^{-1}$ > $g_s$ > 100 mmol m$^{-2}$ s$^{-1}$; severe drought: 100 mmol m$^{-2}$ s$^{-1}$ > $g_s$ > 50 mmol m$^{-2}$ s$^{-1}$; very severe drought: $g_s$ < 50 mmol m$^{-2}$ s$^{-1}$). According to their definition, *Hydrangea* plants under deficit irrigation (experiment one) already experienced moderate drought stress while the standard irrigation was under mild drought stress because an average volumetric substrate moisture content of 26 vol% is rather low [45]. This low substrate moisture content, despite a standard irrigation scheme, can be explained by the overhead irrigation leading to an umbrella effect due to the plants' foliage preventing the irrigation water to reach the substrate. This can be prevented by drip irrigation, used in the second experiment. Here, in the last drying cycle (August 2021), the plants were subjected to severe drought stress. In general, stomatal conductance measured at midday during the first drying cycle in 2021 was half that measured during the last cycle. This first cycle was additionally characterized by a higher light intensity and vapor pressure deficit which explains the lower values [43,47,48].

Stem diameter variations show diel dynamics. Soon after dawn, there is a time delay between the water lost from the plant via leaf transpiration and the water uptake by the roots, causing plants to use water stored in their internal stem reserves, resulting in stem diameter shrinkage. Shortly after noon, the sap flow reaches its daily maximum, and the stem shrinks rapidly. In the afternoon, the sap flow and stem shrinkage both decrease. During the night, when sap flow is the lowest, internal water storage pools are replenished and the stem will swell [49]. When plants are depleted of water and there is not enough water available in the soil/substrate to respond to the evaporative demand of the atmosphere, the maximum daily stem shrinkage increases [49,50]. This pattern is also observed in *Hydrangea*.

Both deficit irrigation and repeated drying cycles reduced the biomass production and branch length of the hydrangeas in this study. Furthermore, in a study by Cameron et al. (2006), the vegetative growth of different woody ornamental species, e.g., *Forsythia*, *Cotinus* was reduced under deficit irrigation while the effect on *Hydrangea macrophylla* at the end of the season was limited. Reduced growth can be a favorable effect for ornamental plants because the combination of shorter shoot lengths/shorter internodes improves compactness and reduces the need for mid-season pruning to become a compact, well-branched plant [51]. In our experiment with deficit irrigation, a significantly higher branching rate was observed in the stressed plants. This was not the case when the hydrangeas were grown under repeated drying and rewetting cycles. Cameron et al. (2006, 2008) also observed no effect of deficit irrigation on the number of shoots and number of formative primary shoots [51,52]. Induced flowering can be a response in many plant species to stressors such as drought, poor nutrition, and light quality. This response is biologically advantageous, especially in plants that produce fertile seeds [53]. For *Hydrangea*, no significant effect of the repeated drying and rewetting cycles on the number of inflorescences was observed, nor on inflorescence development with a higher percentage of inflorescences in a less advanced stage compared with well-irrigated plants. Also found in *Forsythia*, *Cotinus*, and *Hydrangea macrophylla*, the flower number per node was unaffected by the deficit irrigation [51].

### 4.2. Effects of Biostimulants on Hydrangea Grown under Deficit Irrigation or Repeated Drying Cycles

The betaines in seaweed extracts enhance the leaf chlorophyll content [15], which could be due to a reduction in chlorophyll degradation [11]. In grapevine, the negative effect of drought stress on the chlorophyll content was lower in plants treated with the different types of biostimulants compared with the untreated control plants grown under the same stress conditions [54]. In our study, chlorophyll did not appear to be degraded by the presence of drought. On the contrary, deficit irrigation increased the chlorophyll index and REIP (Red Edge Inflection Point) compared with the well-watered No stress Controls. The tested biostimulants did not provide any added value to the deficit irrigation. Seaweed extracts have been reported to increase important bioactive molecular concentrations such as phenolics, flavonoids, and anthocyanins in several crops, such as vegetables and grapevine grown under both optimal and stressed conditions [54–57]. Moreover, in *Calibrachoa* under optimal conditions, increases in phenolic and flavonol contents were found following treatment with a seaweed extract [58]. In this study on *Hydrangea paniculata* under drought, no increase in flavonoid or anthocyanin content, determined by optical sensors, was observed by any of the biostimulants. $F_V/F_M$ was not affected by the deficit irrigation, and biostimulants did not affect this parameter. This was also the case for the treatment of spinach under drought with a seaweed-based biostimulant [59].

Stomatal closure is regulated, among other hormones and mechanisms, by the accumulation of abscisic acid (ABA), which is induced under drought stress [60]. In research on grapevine under drought stress, vines treated with different types of biostimulants, especially a seaweed-based product, accumulated higher levels of ABA compared with the untreated controls to reduce water loss and increase plant drought tolerance [54]. Biostimulants can also act by postponing drought stress, as in the study of Campobenedetto et al. (2021) where a seaweed-based biostimulant reduced the ABA concentration in tomato compared with the untreated control grown under the same mild drought conditions [61]. In other vegetables, such as spinach and broccoli, the application of seaweed-based biostimulants significantly increased stomatal conductance [59,62]. In contrast, the biostimulants in our trials had no significant effect on the stomatal conductance of *Hydrangea paniculata* grown under drought conditions. The *A. nodosum* extract under deficit irrigation slightly increased the stomatal conductance, but this was not linked to a better water-use efficiency as plants had a lower water content at the end of the trial. Treatment with *E. maxima* also showed some effect, but in both cases, they were not significant and not repetitive.

In this research on *Hydrangea paniculata* under water shortage, *A. nodosum* slightly reduced stem shrinkage during drought compared with the control (DS Control), indicating some alleviation of the imposed stress. Previsan S, on the other hand, increased the shrinkage compared with the stressed control (DS control). Top et al. (2023) tested similar seaweed-based products on tomato (*Solanum lycopersicum*) plants under deficit irrigation. For this crop, *A. nodosum* showed no beneficial effect compared with the stressed control. However, another tested *Ascophyllum nodosum* extract (Asco-N2) reduced stem shrinkage in drought-treated tomato plants, similar to *A. nodosum* in *Hydrangea*, and resulted in a similar performance and water uptake as untreated, well-watered control tomato plants [63]. These results might indicate that *A. nodosum*-derived biostimulants can mitigate drought stress to some extent.

The beneficial effects of seaweed extracts on shoot growth and yield were reported in several studies on different crops [9,13], including several ornamentals such as *Calibrachoa*, [58], rose [64], and *Pelargonium* [65]. The effect was mostly dependent on the dose and application method. The tested seaweed extracts in our study had limited effects on the morphological growth parameters and plant quality of hydrangea. *Hydrangea paniculata*, grown under deficit irrigation and treated with *A. nodosum*, had significantly longer branches and more dry weight, but this biostimulant had a rather negative effect on plant growth under repeated drying and wetting cycles. Seaweed extracts induce early

flowering in several crops [9]. The positive effects of a seaweed extract on the flowering and fruit set numbers of eggplant were observed in field conditions [66]. Furthermore, in research on container-grown roses, the application of a seaweed extract increased the flowering [64]. The *A. nodosum* product in our research accelerated flowering, as (significantly) higher numbers of fully developed inflorescences were counted at the end of the growing season. The experimental product based on *S. latissima* and Previsan S also gave similar results. Plants treated with *E. maxima* had significantly fewer branches and thus significantly fewer inflorescences in total. This lower number of branches could be the effect of the biostimulant or of the pruning in June.

## 5. Conclusions

Reduced irrigation resulted in more compact hydrangeas, but this effect is more pronounced under continuous deficit irrigation than under repeated drying and wetting cycles. Although plants were more compact, they also produced less biomass as stomata closed under dry conditions and thus reduced gas exchanges, although photosystem 2 remained intact.

The tested biostimulants in this study had only limited effects on the morphological parameters of *Hydrangea paniculata*, depending on the applied drought stress treatment (deficit irrigation or repeated drying cycles). The *Ascophyllum nodosum* extract positively influenced plant growth under deficit irrigation, and flowering under repeated drying cycles. Flowering could not be assessed under deficit irrigation. This specific biostimulant also slightly reduced stem shrinkage under drought, which might indicate better plant-water relations. The *Ecklonia maxima* extract negatively influenced branching and flowering under repeated drying and wetting cycles, but not under deficit irrigation. From this study, it can be concluded that it is difficult to observe repeated effects under field conditions.

In general, research on biostimulants is complex, as effects seem to depend on many factors such as plant, cultivar, application dose, and method, but also growing conditions which fluctuate widely in the field. This latter aspect increases the challenge of determining the perfect combination between biostimulant, dose, application timing, crop, and growing conditions.

**Supplementary Materials:** The following supporting information can be downloaded at: https://www.mdpi.com/article/10.3390/horticulturae9040509/s1, Table S1: Effect of biostimulant treatments on pigment reflectance indices of *Hydrangea paniculata* grown under deficit irrigation compared with the untreated control (DS Control). Different letters per parameter per measurement day indicate a significant difference at $p \leq 0.05$ (Mean ± SE; $n = 16$), Table S2: Effect of biostimulant treatments on stomatal conductance of *Hydrangea paniculata* grown under deficit irrigation compared with the untreated control (DS Control). Different letters per parameter per measurement day indicate a significant difference at $p \leq 0.05$ (Mean ± SE; $n = 16$), Table S3: Effect of repeated drying and rewetting cycles and biostimulant treatments on pigment reflectance indices of *Hydrangea paniculata* (experiment 2) compared with an untreated control (DS Control) and an untreated control under optimal irrigation (No stress Control). Different letters per parameter per measurement day indicate a significant difference at $p \leq 0.05$ (Mean ± SE; $n = 16$), Figure S1: Examination of the root development of *Hydrangea paniculata* on a relative scale from 1 to 5 after experiment one: 1—almost no visible roots, 2—limited visible roots at the bottom, 3—well-developed roots, but not all around the pot, 4—good root development all around the pot and 5—excellent rooting, Figure S2: Examination of the root development of *Hydrangea paniculata* on a relative scale from 1 to 5 after experiment 2: 1—almost no visible roots, 2—limited visible roots at the side, 3—limited root development at both the side and the bottom, 4—root development all around the pot and limited at the bottom, 5—good root development all around the pot and at the bottom. Plants were grown in bigger containers, Figure S3: Evaluation of development of inflorescences of *Hydrangea paniculata* on a relative scale from 1 to 5 after experiment 2: 1—closed bud, 2—first elongation of the inflorescence with flower clusters still together, 3—second elongation of the inflorescence with the extension of the green flower clusters, 4—flowers open but still green, 5—whitening of the flower.

**Author Contributions:** Conceptualization, E.P., K.S., and M.-C.V.L.; methodology, P.D.C., S.T., E.P., K.S., and M.-C.V.L.; software, K.S. and S.T.; formal analysis, P.D.C. and S.T.; investigation, P.D.C. and S.T.; data curation, P.D.C. and S.T.; writing—original draft preparation, P.D.C; writing—review and editing, P.D.C., S.T., E.P., K.S., and M.-C.V.L.; visualization, P.D.C.; supervision, M.-C.V.L.; project administration, E.P., M.-C.V.L., and K.S.; funding acquisition, E.P., M.-C.V.L., and K.S. All authors have read and agreed to the published version of the manuscript.

**Funding:** This research was funded by Interreg 2 Seas Mers Zeeën, grant number 2S03-029.

**Data Availability Statement:** Data will be available without any reservations by the authors to qualified researchers.

**Acknowledgments:** The author would like to thank Simon Van Kerkhove, Hanne Denaeghel, and Ellen Dams for organizing the trials, their on-site coordination, insights, and ideas about results and also, the entire Bio4safe consortium for the nice collaboration.

**Conflicts of Interest:** The authors declare no conflict of interest.

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
