# Peer review of "Effect of Seaweed-Based Biostimulants on Growth and Development of Hydrangea paniculata under Continuous or Periodic Drought Stress"

_horticulturae, doi:10.3390/horticulturae9040509_

Round 1

Reviewer 1 Report

This article studies the Effect of seesawed based organisms on growth and development of container-grown Hydrangea paniculata under continu. This work is very interesting and has important reference value for researchers in related fields.". Before publishing, several minor issues should be revised.

1Lines 75-79, For the current work plan, it is somewhat simple to write. It is necessary to explain which four types of commercial biologics are used to compare with biostimulant based on seaweed extracts, and why these four types are selected? What is the relationship between these biostimulants? What is the purpose and longer-term value of this job?

2. What does a and b in Table 3 represent? Definition of a and b should be filled in the footer of the table.".

3. Figure 1 should be redone. The picture is too low. Each diagram does not have a title on the horizontal axis. What are the a and b legends in the diagram? You need to define them (a and b). Similar modifications should be made to other drawings.

4. In figure 3, the dates are not standard, 23/6, 24/8, and 25/5. Which year do they represent?

5. For conclusion part, The main conclusions of the paper should be summarized as the conclusion part is too simple.

Author Response

Dear reviewer,

Thank you for the feedback and remarks.

Hereby our comments on your remarks:

  1. The workplan is indeed a bit simple introduced. Now the description of the trial (line 75-79) is extended by the names of the different seaweeds and the reason why they were selected. Also the purpose of the research is mentioned.
  2. The definition of a and b is added to the description of the table, in the same way as for the figures.
  3. All figures are redone and made bigger to increase readability and quality. The designation of the separate figures in figure 1 was changed to a number reference instead of letter reference to avoid confusion with the letters representing significant differences. The different letters (a and b) representing significant differences are defined in the description of the figure.
  4. Figure 3 is redone and is now showing the full date. 
  5. We agree on the fact that the conclusion part is too simple. We extended it with our main conclusions from the research and some future perspectives.

Best regards,

The authors

Reviewer 2 Report

Dear Authors

The manuscript titled ‘Effect of seaweed-based biostimulants on growth and development of container-grown Hydrangea paniculata under continuous or periodic drought stress' is written quite clearly and concisely. The Introduction briefly place the study in a broad context and highlight why it is important. It defines the purpose of the research and its significance. The methods and results are sufficiently described. In the Discussion chapter the authors meticulously and critically referred to the results obtained and used properly selected references. The conclusions correspond to the aim of the research and are supported by the obtained results. I only have some minor comments that need to be addressed.

1.       Line 17 - Research was conducted in 2019 and 2021. Delete the word "consecutive".

2.       Lines 85 and 88 – change L to dm3

3.       Line 215, 234, 269 - I recommend shortening the subsection titles:

3.1.     Effect of deficit irrigation and biostimulants in 2019’’

3.1.2.      Effect of biostimulants

3.2.     Effect of repeated drying and wetting cycles and biostimulants in 2021

4.       Lines 235-252 - Avoid excessive use of values and standard deviations in the text.

5.       References must be verified. The strictly necessary references for the understanding of the article should be cited. Try to reduce references in your article. Avoid using outdated references e.g. 18; 19; 20; 34; 35; 36; 42; 45; 72. 

Author Response

Dear reviewer,

Thank you for the feedback and your remarks.

Hereby our comments on your remarks:

  1. The word 'consecutive' is omitted as it is indeed unnecessary.
  2. We changed the unit of the container volume to dm³.
  3. The subtitles are shortened to make them more readable.
  4. There are indeed a lot of values in this part of the text (lines 235-253). We decided to omit the less interesting values that are linked to the supplementary files or describe the differences between 2 treatments in %. (Lines 249, 251, 252, 253-254).
  5. In the case of references 18 (Rouse et al., 1974), 19 (Lichtenthaler, 1996) and 20 (Gitelson et al., 1994), we are referring to the original papers where the reflectance index is mentioned for the first time. We consider these studies as fundamental in that research domain.

    Reference 34 refers to the first time the Scheirer-Ray-Hare test is defined.

    Reference 35 (Turner, 1986) is omitted as this paper focuses on the research between 1980-1990.

    Reference 36 (Turner, 1991), we consider as a fundamental review paper on the environmental effects on the stomatal conductance.

    References 41 and 42 are considered as specific basic research on the effect of environmental factors on stomatal conductivity, together with reference 35.

    Reference 45 (Gitelson et al., 1996) is considered as basic research on red edge reflection and inflection point.

    Reference 72 is omitted as there is also referred to other papers about effect of biostimulants on flowering.

    The other references were also verified for their necessity. Other papers that are left out are reference 4 (Li et al., 2009), 51 (Taiz and Zeiger, 2003), 57 (Cameron et al., 1999), 59 (Wright, 1977) and 64 (Kaluzewicz et al., 2018).

Best regards,

The authors

Reviewer 3 Report

I have reviewed this manuscript with interest. Although this manuscript contains some useful information, the authors should consider revising the description of the results extensively and their discussion. The conclusions should also be modified.

The can find my specific comments in the PDF version of the manuscript (See file attached)

Author Response

Dear reviewer,

Thank you for your remarks and feedback.

Hereby our comments to you remarks:

Remark

Reply

1.       Line 3: I think there is no need to specify this in the title.

The term ‘container-grown’ is left out and added to the abstract instead.

2.       Line 82: what does PCS stands for?

The abbreviation can be explained by the full Dutch name of the company, this is added between parentheses (new line, 87-88)

3.       Line 83: Specify the latitude and longitude of these coordinates.

Latitude and longitude are added (new line 88).

4.       Line 84: do you mean root cuttings? Please confirm.

Rooted stem cuttings of rooted plug plants are used, as these cuttings are not used to propagate plants. (new line 90).

5.       Line 88: Kindly specify the shape of the containers (circular, squared, etc.).

Circular containers are the most used type in Europe. That is the reason why we did not describe the shape. (new line 94)

6.       Line 90: are they many? If yes, consider adding the plural form.

There is more than one trace element present. The plural form is used. (new line 96).

7.     Line 91: in a greenhouse should be the correct indication.

‘The’ is replaced by ‘a’. (new line 97)

8.     Line 91: use the minus signe to indicate the range, not the dash symbol. Check throughout the manuscript.

We adjusted the use of the dash symbol through the whole manuscript.

9.     Line 93, indicate how many plants are planted and evaluated.

The numbers are mentioned in the descriptions of the separate trials in the following sections. (Line 117-120 and line 142-144 of the revised manuscript).

10.   Line 92: replace with a minus sign.

Adjusted through whole manuscript.

11.   Line 94: replace with a minus sign.

Adjusted through whole manuscript.

12.   Line 97: It will be more informative if the authors specify the days after planting or sowing or transplanting while keeping the date in parentheses. This applies to Experiment 2 as well.

We agree on this as the day after planting is more informative. The information is added, also for experiment 2. (new lines: 103-104 and 128-129).

13.   Line 98: replace the dash with a minus sign.

Adjusted through whole manuscript.

14.   Line 99: What is the basis for using 80% reduction in irrigation? Does it come from an previous experiment or screening? Kindly provide the rational.

A 10 % reduction was tested during a preliminary trial and had minor effects on plant growth and stress level,  so no effects of the biostimulants were present/visible. Therefore, we reduced the irrigation with 20 % in 2019 to create a mild drought stress. With 20 %, the reduction was also not too strong. This rational is also added to the manuscript (new lines 106-107).

15.   Line 100: At which soil moisture the authors considered drought stress was induced? Specify in terms of water potential or soi moisture percentage. What do you mean by 80% irrigation? Is it 20% reduction in water supply compared to the control? Is that enough to consider drought stress conditions? Explain and describ clearly the experimental method for drought induction, assessement and evaluation (Check where to include these details. May be in 2.4

The specific thresholds when drought stress is expected are discussed in the discussion section.

Indeed, 80 % irrigation corresponds to a reduction of 20%, this specific description is added to the manuscript (line 106-108).

The substrate moisture content was monitored two-weekly for the first trial, this is described in section 2.4 (line 151-153).

16.   Line 101: How old were these plants before the reduced water supply and after.

The information is added in new lines: 103-104 and 128-129.

17.   Line 107-109: This description should come first to introduce the experimental design...Consider specifying the total number of plants evaluated

Together with the introduction of the treatments the number of treatments is mentioned, so that the summary of the experimental design at the end of the paragraph is more clear.

The total number of plants is added for both trials.

18.   Line 183: replace with a minus sign.

Adjusted through the whole manuscript.

19.   Line 217: The authors should improve the description of the results. They should start by giving some context of the exepriment for the parameter before they indicate the treatment effect.

i.e. To investigate "x" treatment effect on "y" material, we measured "z" parameter". Results show that...

Note: this comments applies to all results.

In addition, the result title should reflect the main finding of the study for each parameter observed. Giving a general title do not help understand the study and is less informative

The reason why we used different sensors and methods is described in the subsection ‘Materials and Methods’, but is repeated in the Result part to give the results some context and make it more clear and understandable.

We decided to rephrase and shorten the titles to make them better readable based on the comments on reviewer 2. We did not include results in the titles, as they would become too long and less readable in this way.

20.   Line 217-218: The authors should maintain a sequencial description of their results that give a clear picture of the results in detail.

The results of both experiments are most of the time described in the same sequential way (Substrate-related parameters: substrate moisture content and EC, followed by the plant-related parameters: Chlorophyll and flavonol index, reflectance indices (REIP, NDVI, Ctr2, Lic1, ARI1), Fv/Fm, Stomatal conductance, branch length and number of branches, FW and DW, root development). Moments of deviation from this order were adjusted, mainly for experiment 2, such that in the revised manuscript the order is the same for both experiments.

21.   Line 221: This statement could be change to: "Figure or Table x shows that chlorophyll content was higher in leaves of....but low in other plant organs.

I am suggesting to rephrase the description and put the results as a story

Some sentences are rephrased throughout the result section to put the results more smoothly in a story.

22.   Table 3 the lettering indicating the statistical difference is general put close to the value not far from the value

Table 3 is adjusted and lettering indicating the statistical difference is put more close to the value. 

23.   Line 234: All results titles should be rephrased to proved more informative titles

We decided to rephrase and shorten the titles to make them better readable based on the comments on reviewer 2. We did not include results in the titles, as they would become too long and less readable in this way.

24.   Line 237: This section can be improve to provide more insight into the results. The use of wordings such as "influence" does not help understand the extend of the effect. The authors should use expressions such as ...increase by x%... or decreased by x%...

The increase or decrease is described in percentages as it is indeed better readable and understandable compared to the raw numbers.

An attempt was made to avoid terms like ‘influence’ most of the time to provide more insights.

25.   Line 379: This title is similar to that presented in the result section. The authors should consider giving titles that reflect the best of their results.

The entire discussion section ressembles the results and should be revised extensively. Here, the authors should discuss their findings in a broad context.

The title is rephrased and shortened. Reflection of the results would result in a too long title.

The discussion is revised and findings are more defined in a broad context, although we did not only focus on the application of biostimulants on ornamentals but also on vegetables and arable crops. 

26.   Line 465-466: provide a more meaningful subtitle that show the finding not repeating the results title

The subtitles are rephrased and a bit shortened. 

27.   Line 542: This section needs improvement and try to put the findings into perspective.

We agree on this. The conclusion section is extended with more findings of the experiments and are more put into perspective. 

Best regards,

The authors

Reviewer 4 Report

Dear Authors,

The reviewed paper is an example of a scientific publication well prepared by the authors for publication. My comments are of an editorial nature only.

1. under the tables (in the main text as well as in Supplementary Materials) please explain the abbreviations used in them

2. the notation of units of measurement in the main text of the manuscript, in Figure 3 and in Supplementary Materials should follow the SI system, e.g. instead of mmol.m-2 .s-1 it should be mmol۰m-2۰s-1.

3. figure 1f - remove the double bracket.

4. Figure 3 could be enlarged slightly - it is not very readable. Similarly the description of both axes and the legend on the graph

5. References: according to journal requirements, Latin names of species should be written in italics; the doi notation should be standardised.

Kind regards,

Reviewer

Author Response

Dear reviewer,

Thank you for your feedback and remarks.

Hereby our comments on your remarks:

  1. The explanations of all abbreviations are added in the text, table or figures the first time they are used. In figure 3, abbreviations are explained. Also in the main text, we added the full names of the abbreviations with the start of a new section of the paper.
  2. Notation of units is changed following the SI system in the main text, Figure 3 and TableS2.
  3. Double bracket is needed in the formula in figure 1f, so the outer ones are replaced by [], to make it better readable.
  4. All figures, including graph 3 were enlarged.
  5. The reference list is controlled on the fact that latin names are written in italics and doi notation is standardized.

Best regards,

The authors

Round 2

Reviewer 3 Report

Thank you for making the necessary changes to the manuscript. The manuscript has been significantly improved by the authors.